# Have Chest Imaging Habits Changed in the Emergency Department after the Pandemic?

Cüneyt Arıkan [1,*] , Ejder Saylav Bora [2] , Efe Kanter [3] and Fatma Nur Karaarslan [1]

1  Department of Emergency Medicine, Soma State Hospital, Manisa 45500, Turkey; f.nurkaraarslan@gmail.com
2  Department of Emergency Medicine, Izmir Ataturk Training and Research Hospital, Izmir 35150, Turkey; saylavbora@hotmail.com
3  Department of Emergency Medicine, Izmir Kâtip Çelebi University, Izmir 35620, Turkey; efekanter@hotmail.com
*  Correspondence: dr.carikan@gmail.com

**Abstract:** The rate of patients undergoing tomography in the emergency department has increased in the last two decades. In the last few years, there has been a more significant increase due to the effects of the pandemic. This study aimed to determine the rate of patients who underwent chest imaging in the emergency department, the preferred imaging method, and the demographic characteristics of the patients undergoing imaging during the pre-pandemic and post-pandemic periods. This retrospective cross-sectional study included patients admitted to the emergency department between January 2019 and March 2023. The number of female, male, and total emergency admissions, the rate of patients who underwent chest X-ray (CXR) and chest computed tomography (CCT), and the age and gender distribution of the cases who underwent chest imaging were compared according to the pre-pandemic (January 2019–February 2020), pandemic (March 2020–March 2022), and post-pandemic (April 2022–March 2023) periods. Total emergency admissions were similar in the pre-pandemic and post-pandemic periods (pre-pandemic period: 21,984 ± 2087; post-pandemic period: 22,732 ± 1701). Compared to the pre-pandemic period, the CCT rate increased (pre-pandemic period: 4.9 ± 0.9, post-pandemic period: 7.46 ± 1.2), and the CXR rate decreased (pre-pandemic period: 16.6 ± 1.7%, post-pandemic period: 13.3 ± 1.9%) in the post-pandemic period ($p < 0.001$). The mean age of patients who underwent chest imaging (CXR; Pre-pandemic period: 56.6 ± 1.1 years; post-pandemic period: 53.3 ± 5.6 years. CCT; Pre-pandemic period: 68.5 ± 1.7 years; post-pandemic period: 61 ± 4.0 years) in the post-pandemic period was lower than in the pre-pandemic period ($p < 0.001$). Chest imaging preferences in the emergency department have changed during the post-pandemic period. In the post-pandemic period, while younger patients underwent chest imaging in the emergency department, CCT was preferred, and the rate of CXR decreased. It is alarming for public health that patients are exposed to higher doses of radiation at a younger age.

**Keywords:** chest imaging; chest X-ray; pandemic; chest computed tomography





## 1. Introduction

Chest radiography (CXR) is usually the first and most commonly used chest imaging method in the emergency department (ED) [1]. Reasons for this include the ease with which CXR can be performed (including bedside studies in critically ill patients), rapid interpretation, less radiation exposure to the patient, and lower costs compared with computed tomography (CT) scans [2,3]. While it has significant limitations compared to CT, such as missing small lesions, failure to evaluate vascular structures, and superposition, with an appropriate clinical approach, the diagnosis rate for critical diseases (such as left ventricular failure, pneumonia, pleural effusion, or rib fracture) in the emergency department is acceptably high [1]. Past studies have reported that the margin of error may be between 10% and 15% despite the systematic approach [4].

In CT, a focused X-ray beam is directed at the patient, and after focusing on the patient, the beam is rapidly rotated around the patient's body. The generated signals are then evaluated by a computer housed within the apparatus, which ultimately results in the creation of cross-sectional photographs, also referred to as slices. CT scans allow for cross-sectional imaging and improved visualization of abnormalities compared to CXRs. This is achieved by enhancing contrast and eliminating the superimposition of structures due to tomographic sectioning, which can be conducted in virtually any plane [1,5]. Chest computed tomography (CCT) is increasingly used in ED because it allows the evaluation of vascular structures, airways, mediastinum, and the heart in addition to the lungs [5].

Before the COVID-19 pandemic, ED visits and using CT in the ED increased at similar rates yearly [6–8]. However, this situation changed during the pandemic period. The SARS-CoV-2 (Severe Acute Respiratory Syndrome Corona Virus-2) virus, which spread rapidly after it first appeared and became a significant public health problem globally, caused human deaths, mainly due to lung failure [9,10]. The World Health Organization (WHO) declared an "International Public Health Emergency" on 30 January 2020 and then a global "Pandemic" on 11 March 2020 [9,11]. In order to reduce the spread of the virus and control the disease worldwide, many measures, especially quarantine, were taken, and restrictions were imposed on daily life. With the effect of the vaccines developed later, the spread of the virus was brought under control, and its mortality decreased [12]. Bans and restrictions were lifted following decreased cases worldwide, ED admissions, hospitalizations, and mortality. The process of returning to everyday life finally began, and the WHO declared that the pandemic was over [13].

Although the pandemic is over, its effects on healthcare systems may continue. During the pandemic, the excessive load of critically ill patients affected all clinics, especially emergency services. One of the branches most affected was radiology. Due to the lung involvement of the virus, the number of patients undergoing thorax imaging increased significantly. During the pandemic, there was a significant increase in the number of patients who underwent CT imaging [14–16].

With the restrictions and stay-at-home measures during the pandemic, there was a significant decrease in admissions to hospitals and EDs [17,18]. However, despite decreased ED visits and the fact that CXR helps diagnose and prognosis COVID-19, the number of CT imaging scans per patient increased considerably [14,19,20]. This increase is thought to be due to the increased diagnostic accuracy of CT imaging for COVID-19, increased risk of thromboembolism, increased rate of patients admitted by ambulance, and a higher rate of critical patient admissions to EDs [20–23]. In addition to changes in imaging preference and number during the pandemic, changes in the demographic characteristics of patients undergoing imaging were also reported. The mean age of patients undergoing CCT decreased by approximately 15 years, and the radiation exposure of the younger population increased significantly [15].

The COVID-19 pandemic was brought under control, and the number of reported cases began to decline, so the quarantine restrictions were lifted, and people worldwide began returning to their everyday lives. Almost all emergency department visits and hospitalizations attributed to a possible case of COVID-19 have been eliminated. Because of this, the number of patients admitted to the ED has increased and is now at the same level. However, it has yet to be determined how the COVID-19 pandemic will affect the utilization of resources in the emergency department and how it will influence the imaging methods used by ED staff. It is not yet known whether the changes in chest imaging methods seen in emergency departments during the pandemic period will continue in the post-pandemic period.

This study will not only determine whether chest imaging habits have changed in the emergency department. However, it will also try to show the transmission of imaging habits between physicians in their professional working lives, starting with students whose face-to-face education processes have been interrupted during the pandemic.

The purpose of this research was to determine whether or not there was a shift in the preferences for chest imaging in the ED, as well as the demographic characteristics of patients who underwent imaging before and after the pandemic.

## 2. Materials and Methods

### 2.1. Study Design and Settings

This was a single-center retrospective cross-sectional study carried out in the emergency medicine clinic of a tertiary care hospital in a metropolitan area with a population of roughly 4.5 million. Approximately 250,000 patient visits are made each year to the hospital's emergency department (ED), which has a total bed capacity of one thousand. The Izmir Katip Celebi University Non-Interventional Clinical Research Ethics Committee approved the study (Decision No: 0129, Date: 23 March 2023). This was carried out so that the research could proceed. Because the study was conducted retrospectively, separate patient consent was not collected in addition to the ethics committee's approval.

### 2.2. Study Population

Patients admitted to the adult emergency department of the hospital between January 2019 and March 2023 were included in the study. Patients with missing data in their files and patients who were referred from another hospital and had imaging in that hospital were excluded from the study by checking the National Health Bank e-nabız (an application where citizens and healthcare professionals can access health data collected from healthcare institutions via internet and mobile devices), which can only be used by physicians.

### 2.3. Data Collection and Processing

The number of female, male, and total number of emergency admissions, the number of patients undergoing CXR, the number of patients undergoing CCT, and the demographic characteristics (age and gender) of patients undergoing chest imaging were obtained from the electronic medical records of the hospital monthly and the data were digitized. The number of CXR and CCT scans performed per 100 patients and the mean age of the patients were calculated from the data obtained. The period during which the study data were obtained was divided into three parts: the pre-pandemic period (January 2019–February 2020), the pandemic period (March 2020–March 2022), and the post-pandemic period (April 2022–March 2023). Within the period covered by the study, the period before the WHO declared the pandemic was defined as the pre-pandemic period. A period longer than two years, when COVID-19 restriction measures were intensively implemented in Turkey, and the total number of cases and the number of patients hospitalized and in intensive care were highest in Turkey and in the center where the study was conducted, was defined as the pandemic period. During the period defined as the pandemic period, approximately one-third of the total hospital bed capacity was used only for COVID-19 patients. With the COVID-19 epidemic being controlled, the period in which admissions to the ED and hospitalizations due to suspicion of COVID-19 almost entirely ended, quarantine measures were lifted, and society returned to everyday life was defined as the post-pandemic period.

### 2.4. Variables

Comparisons were made between the pre-pandemic, pandemic, and post-pandemic periods regarding the number of CXRs and CCTs performed per 100 admissions to ED and the age and gender distribution of the patients who underwent imaging.

Statistical Package for the Social Sciences (SPSS) 25.0 (Chicago, IL, USA) was used for data analysis. The Shapiro–Wilk test and histograms were used to evaluate normal distribution. Normally distributed data were presented as mean $\pm$ standard deviation, and non-normally distributed data were presented as median and interquartile range. Frequencies were used for categorical data. The changes of the variables investigated according to periods were evaluated by repeated measure ANOVA (ANOVA) tests. Bonferroni correc-

tion was presented in rmANOVA post hoc tests. In all analyses, $p < 0.05$ was considered statistically significant.

## 3. Results

During the study period, a total of 986,750 admissions were evaluated. There were an average of $19,348 \pm 4200$ emergency admissions per month. Among these admissions, 506,566 were male, and 480,184 were female. The admission rate in males was significantly higher than in females ($p < 0.001$). The total number of CXRs taken was 136,632, and the total number of CCT scans was 94,545. For every 100 emergency admissions, an average of $13.8 \pm 2.7$ CXR and $10.9 \pm 7.6$ CCT scans were performed. The mean age of the patients who underwent CXR was $53.7 \pm 3.4$ years, and those who underwent CT scans were $62.4 \pm 5.2$ years. CXR imaging was performed in $15.3 \pm 2.7\%$ and CCT imaging in $11.9 \pm 7.8\%$ of male and $12.3 \pm 2.8\%$ and $9.9 \pm 7.4\%$ of female patients, respectively.

Total emergency admissions, demographic characteristics of the cases, and chest imaging methods were compared according to pre-pandemic, pandemic, and post-pandemic periods. Emergency admissions changed significantly before, during, and after the pandemic ($p < 0.001$, ANOVA). During the study period, it was observed that there were a total of 307,781 admissions in the pre-pandemic period, 406,189 admissions during the pandemic period, and 272,780 admissions in the post-pandemic period. In the pre-pandemic period, the average monthly admission rate was $21,984 \pm 2087$; during the pandemic period, it was $16,248 \pm 3643$; in the post-pandemic period, it was $22,732 \pm 1701$. The pandemic was observed to significantly decrease the number of patients admitted to the ED (F = 39.277; $p < 0.001$; partial $\eta^2 = 0.887$). According to post hoc comparisons with Bonferroni correction, while there was no significant difference between total emergency admissions before and after the pandemic ($p = 0.557$), it was significantly lower during the pandemic period compared to both periods ($p < 0.001$). The change in the number of ED visits by month is shown in Figure 1.

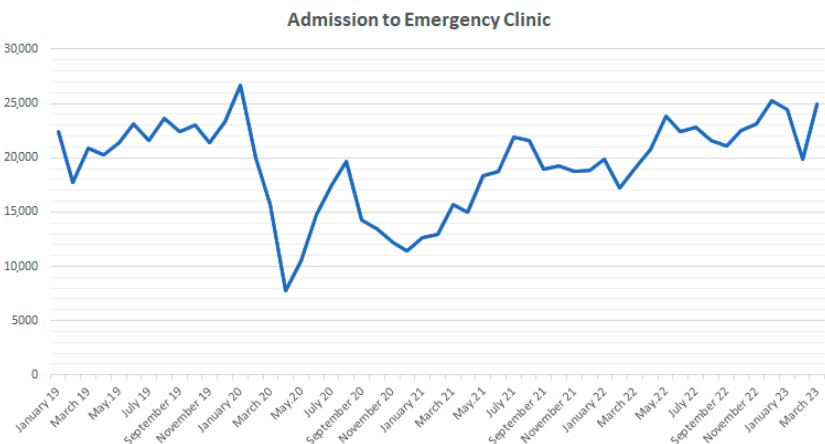

**Figure 1.** Admission to emergency clinic.

The number of male admissions during the pre-pandemic period averaged $11,574 \pm 292$ per month; during the pandemic period, it averaged $8080 \pm 409$ per month, and in the post-pandemic period, it averaged $11,878 \pm 272$ per month. For female admissions, during the pre-pandemic period, it averaged $10,411 \pm 270$ per month; during the pandemic period, it averaged $8162 \pm 324$ per month; in the post-pandemic period, it averaged $10,853 \pm 229$ per month. The rate of emergency admissions by gender varied significantly between the periods before, during, and after the pandemic ($p < 0.001$, ANOVA). The pandemic was observed to increase the proportion of women significantly admitted to the ED (F = 69.455; $p < 0.001$; partial $\eta^2 = 0.863$). According to post hoc comparisons with Bonferroni correction, the percentage of women increased significantly during the pandemic compared to both

before and after the pandemic ($p < 0.001$ for both). However, there was no significant difference in the percentage of women before and after the pandemic ($p = 0.125$).

Pre-pandemic, the average age of patients who underwent CXR was $56.6 \pm 1.1$ years; during the pandemic period, it was $52.3 \pm 1.7$ years, and in the post-pandemic period, it was $53.3 \pm 5.6$ years. The mean age of patients undergoing CXR varied significantly between the pre-pandemic, post-pandemic, and post-pandemic periods ($p < 0.001$, ANOVA). The mean age of patients who underwent CXR during the pandemic decreased significantly (F = 5.053; $p < 0.044$; partial $\eta^2$: 0.315). According to post hoc comparisons with the Bonferroni correction presented, the mean age of the patients who underwent CXR during the pandemic period compared to the pre-pandemic period decreased significantly ($p < 0.001$), and there was no significant change in the post-pandemic period compared to the pandemic period ($p = 0.950$). However, the mean age of the patients who underwent CXR did not change in the post-pandemic period compared to the pre-pandemic period ($p = 0.270$).

Pre-pandemic, the average age of patients who underwent CCT scans was $68.5 \pm 1.7$; during the pandemic period, it was $59.6 \pm 3.9$; in the post-pandemic period, it was $61 \pm 4$. When the mean age of the patients who underwent CCT was examined, it varied significantly between the pre-pandemic, during, and post-pandemic periods ($p < 0.001$, ANOVA). The mean age of patients who underwent CCT during the pandemic decreased significantly (F = 37,352; $p < 0.001$; partial $\eta^2 = 0.773$). According to post hoc comparisons with the Bonferroni correction presented, the mean age of the patients who underwent CCT during the pandemic period compared to the pre-pandemic period was significantly lower ($p < 0.001$), and it increased again in the post-pandemic period compared to the pandemic period ($p < 0.001$). However, the mean age of the patients who underwent CCT in the post-pandemic period compared to the pre-pandemic period was significantly lower ($p < 0.001$). The change in the mean age of the patients screened by months is shown in Figure 2.

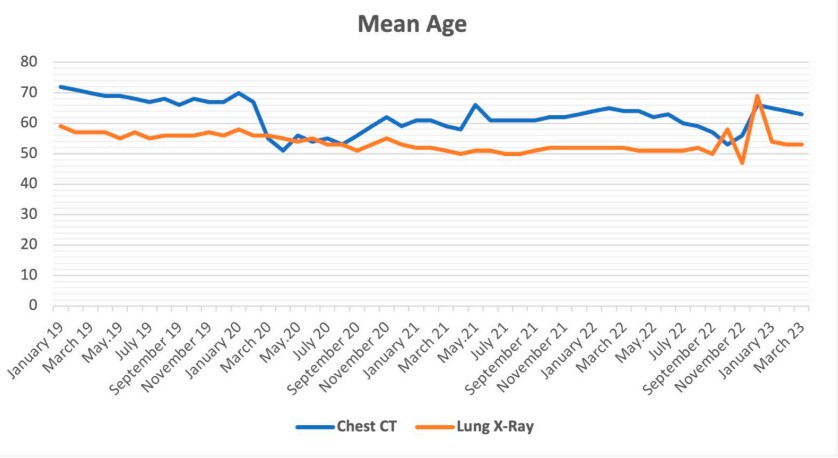

**Figure 2.** Mean age.

During the pre-pandemic period, the rate of CXR performed per 100 admissions to the ED averaged $16.6 \pm 1.7$. However, this rate decreased to an average of $12.4 \pm 2.1$ during the pandemic. It slightly increased in the following post-pandemic period, averaging $13.3 \pm 1.9$ per 100 admissions to the ED. The pandemic significantly decreased the CXR rate (F = 57.344; $p < 0.001$; partial $\eta^2 = 0.839$; ANOVA). According to post hoc comparisons with Bonferroni correction, the CXR rate decreased significantly during the pandemic compared to the pre-pandemic period ($p < 0.001$). It increased significantly in the post-pandemic period compared to the pandemic period ($p = 0.026$). However, it remained significantly lower after the pandemic than before ($p < 0.001$). In both gender-specific analyses, it was observed that the CXR rate was significantly lower in both females and males in the post-pandemic period compared to the pre-pandemic period (Female: F = 49.559; $p < 0.001$; partial $\eta^2 = 0.818$; rmANOVA, Male: F = 31.810; $p < 0.001$; partial $\eta^2 = 0.743$; rmANOVA).

In the pre-pandemic period, an average rate of $4.9 \pm 0.9$ CCT scans were performed per 100 admissions to ED. During the pandemic, this rate increased to $15.9 \pm 8.1$; in the post-pandemic period, it was an average of $7.46 \pm 1.2$. The pandemic significantly increased the CCT rate (F = 62.760; $p < 0.001$; partial $\eta^2 = 0.865$; ANOVA). Bonferroni's corrected post hoc comparisons showed that the CCT rate increased during the pandemic compared to the pre-pandemic period ($p < 0.001$). Although there was a significant decrease in the post-pandemic period compared to the pandemic period ($p < 0.001$), it was still significantly higher than the pre-pandemic period ($p < 0.001$). CXR and CCT rates by month are shown in Figure 3. In females, it was observed that during the pre-pandemic period, the rate of CXR performed per 100 admissions to ED was $15.3 \pm 1.7$; during the pandemic period, it was $10.9 \pm 2.3$; in the post-pandemic period, it was $11.6 \pm 2.1$. For males, during the pre-pandemic period, the rate was $18.1 \pm 1.8$; during the pandemic period, it was $13.8 \pm 2.1$; in the post-pandemic period, it was $15.2 \pm 2.0$. In females, during the pre-pandemic period, the rate of CCT scans performed per 100 admissions to ED was $4.8 \pm 1.0$; during the pandemic period, it was $14.8 \pm 7.9$; in the post-pandemic period, it was $5.7 \pm 1.0$. For males, during the pre-pandemic period, the rate was $5.0 \pm 0.9$; during the pandemic period, it was $17.0 \pm 8.1$; in the post-pandemic period, it was $9.42 \pm 1.7$. In both gender-specific analyses, the CCT rate increased significantly in both females and males in the post-pandemic period compared to the pre-pandemic period (Female: F = 55.619; $p < 0.001$; partial $\eta^2 = 0.835$; rmANOVA, Male: F = 56.647; $p < 0.001$; partial $\eta^2 = 0.837$; rmANOVA). The number of cases according to time intervals, demographic characteristics of the cases, and chest imaging methods are presented in detail in Table 1.

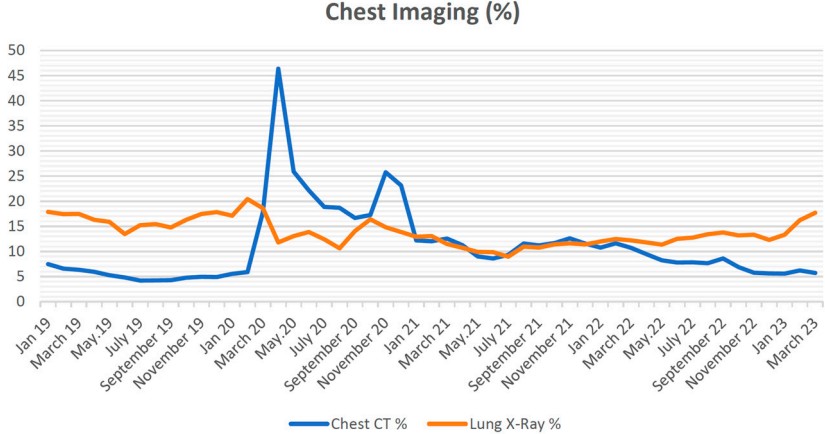

**Figure 3.** Chest imaging (%).

**Table 1.** Number of cases, demographic characteristics of cases, and changes in chest imaging methods over time.

|  | Pre-Pandemic (Mean $\pm$ SD [1]) | Pandemic (Mean $\pm$ SD) | Post-Pandemic (Mean $\pm$ SD) | *p*-Value * |
|---|---|---|---|---|
| **Total Number of Patients** | 21,984 $\pm$ 2087 | 16,248 $\pm$ 3643 | 22,732 $\pm$ 1701 | <0.001 |
| Female (%) | 47.3 $\pm$ 0.2 | 50.6 $\pm$ 0.4 | 47.8 $\pm$ 0.2 | <0.001 |
| Male (%) | 52.7 $\pm$ 0.2 | 49.4 $\pm$ 0.4 | 52.2 $\pm$ 0.2 | <0.001 |
| CXR [2] Mean Age | 56.6 $\pm$ 1.1 | 52.3 $\pm$ 1.7 | 53.3 $\pm$ 5.6 | <0.001 |
| CCT [3] Mean Age | 68.5 $\pm$ 1.7 | 59.6 $\pm$ 3.9 | 61.0 $\pm$ 4.0 | <0.001 |
| CXR (%) | 16.6 $\pm$ 1.7 | 12.4 $\pm$ 2.1 | 13.3 $\pm$ 1.9 | <0.001 |
| CCT (%) | 4.9 $\pm$ 0.9 | 15.9 $\pm$ 8.1 | 7.46 $\pm$ 1.2 | <0.001 |
| CXR Female (%) | 15.3 $\pm$ 1.7 | 10.9 $\pm$ 2.3 | 11.6 $\pm$ 2.1 | <0.001 |
| CXR Male (%) | 18.1 $\pm$ 1.8 | 13.8 $\pm$ 2.1 | 15.2 $\pm$ 2.0 | <0.001 |
| CCT Female (%) | 4.8 $\pm$ 1.0 | 14.8 $\pm$ 7.9 | 5.7 $\pm$ 1.0 | <0.001 |
| CCT Male (%) | 5.0 $\pm$ 0.9 | 17.0 $\pm$ 8.1 | 9.42 $\pm$ 1.7 | <0.001 |

[1] SD: standart deviation, [2] CXR: chest X-ray, [3] CCT: chest computed tomography. * Data obtained from repeated measures ANOVA test.

## 4. Discussion

The COVID-19 pandemic has already occurred and is one of the most important events affecting the world this century. The adverse effects of the pandemic were deeply felt in many areas, especially the healthcare system. One of the branches most affected by the intensity and complexity of hospitals was the ED. Although the number of admissions to the ED has decreased, the change in the patient profile and the measures taken to prevent virus transmission have led to changes in the routine functioning and patient management of EDs. One of the most noticeable changes is the radiological imaging methods performed in the ED. During the pandemic, a significant increase in patients undergoing CCT was reported [14–16]. This increase is also evident in our study. There are already many studies showing the changes during the pandemic period. However, the difference in this study is whether these changes are reversed in the post-pandemic period. In the post-pandemic period, when virus transmission and its impact on the world decreased significantly, quarantine measures were lifted, ED visits due to suspicion of COVID-19 and hospitalizations almost wholly disappeared, hospitals and EDs returned to their routine functioning, and the rate of CCT performed in the ED decreased compared to the rate of CCT during the pandemic period, but all still remain higher than before the pandemic. In addition, more younger patients are imaged in the ED compared to the pre-pandemic period.

CCT was recommended as an appropriate approach for clinicians in the ED when deciding on hospitalization or discharge due to its rapid results compared to PCR testing, easy accessibility, and reliability for diagnosing pneumonia during the pandemic [21,24,25]. CCT was recommended as an appropriate approach due to its rapid results compared to PCR testing, easy accessibility, and reliability for diagnosing pneumonia during the pandemic. During the pandemic, this was one of the most significant reasons for the rise in the utilization of CCT in emergency departments. Nevertheless, the results of our research demonstrated that, compared to the time before the pandemic, there has been an ongoing rise in the utilization of CCT in emergency departments.

The effects of this clinical approach on ED staff during the pandemic may be the main reason for the high rate of CCT in the post-pandemic period. Although these results for the pandemic are understandable and similar to previous studies, continuing these effects in the post-pandemic period worries the healthcare system. Reversing these effects should be one of the main goals of emergency services. Otherwise, the negative consequences of increasing radiation exposure will be inevitable. Being more careful about indications for CCT in the ED is one of the critical points to prevent these adverse effects [15]. In order to prevent excessive and unnecessary use of CCT, it is recommended to create a carefully considered differential diagnosis list based on detailed history and physical examination and evaluate CXR systematically [1]. Another suggestion on this subject is to encourage the increased use of lung ultrasound in the clinic as an alternative to CCT. Much evidence regarding lung ultrasound's value has been reported in the literature [26–28].

Compared to before the pandemic, the percentage of patients who had CXR imaging carried out during the post-pandemic period is lower than during the pre-pandemic period. This is in addition to the rise in the incidence of CCT. According to these findings, CXR is gradually being replaced by CCT in the emergency department due to the pandemic. Compared to the past, this results in higher radiation exposure and increased costs in the emergency department.

Another significant effect of the clinical approach during the pandemic has been the mean age of patients undergoing imaging. According to the results of our study, CCT imaging is performed in younger patients in the post-pandemic period compared to before the pandemic. Previous studies reported that the average age of patients who underwent CCT during the pandemic decreased by ten years or more compared to the pre-pandemic period [15,16]. However, the fact that this effect continues in the post-pandemic period is one of the crucial points that is worrying for the future. Patients are now exposed to radiation at younger ages. The adverse effects of single exposure to high doses or

continuous exposure to low doses of radiation on the human body are well known [29]. Long-term low-dose radiation has been reported to cause thyroid cancer, lung cancer, and leukemia [30]. This research was conducted in the ED, where only adult patients were admitted, and striking results were obtained regarding radiation exposure. This situation, closely related to public health, is especially important for the growing generation [31]. It has been reported that, just like adults, patients in the child and adolescent age groups were also exposed to more radiation during the pandemic period [15]. It has yet to be discovered what the current situation in pediatric clinics is in the post-pandemic period. However, with new research to be conducted shortly, current data in pediatric clinics may be revealed.

A recent study conducted an evaluation of the benefits and limitations associated with digital tomosynthesis (DTS) in comparison to direct radiography and CT, mostly because of its reduced radiation exposure [31]. While ongoing research is being conducted to further substantiate its effectiveness, preliminary findings suggest that dual-energy thoracic radiography (DTS) could potentially serve as a viable alternative imaging modality to chest X-ray and CT scans for monitoring pulmonary changes, particularly ground-glass opacities (GGOs) and "fibrotic-like" interstitial changes. In fact, DTS may even supplant these traditional methods in numerous cases, thereby enhancing the clinical workflow by expanding access to CT imaging, lowering expenses, and prioritizing patient safety [32–34].

In order to determine and control radiation exposure, recording the average radiation dose to which the patient is exposed in the database system after each tomography scan will be useful for follow-up of the patients. In this way, excessive radiation exposure can be prevented by drawing the attention of both healthcare professionals and patients to this issue.

It is known that COVID-19 affects men more, and the number of men diagnosed with COVID-19 is higher than women [20,21,35]. However, in our study, when the total number of patients admitted to the hospital was compared by gender, there was no significant difference before the pandemic, during the pandemic period, and after the pandemic. In addition, it is observed that the ratio of patients who underwent CXR and CCT imaging before and after the pandemic was similar in both sexes. Many studies also report no difference in the rate of men and women who underwent CCT when comparing the pre-pandemic and pandemic periods [15,16,36]. The study results are similar to studies in the literature in this respect.

*Limitations*

The research has some limitations. The most important one of these limitations is that it was conducted on a retrospective and single-center study that needs to be expanded into multi-institutional joint research. Therefore, these results may not represent the entire healthcare system and cannot be generalized to the whole population. An additional constraint of the research was that, for the study, patients admitted to the emergency department were not categorized based on clinical diagnoses, complaints, symptoms, or imaging indications.

## 5. Conclusions

This study indicates that the use of CCT in the emergency department increased in the post-pandemic period compared to the pre-pandemic period. The increase in the use of CCT in the emergency department during the pandemic period and the decrease in the average age of patients undergoing CCT continue in the post-pandemic period, although there is some improvement. In addition, patients in the emergency department are undergoing CCT and CXR scans at a younger age compared to the past. The increase in CT imaging and radiation exposure at a younger age may have short- and long-term adverse effects on patients and the healthcare system. The most important principle of our medical education regarding CT indications in the emergency department is that we should act on the principle of "first do no harm" and protect our current algorithms in terms of

public health. However, the use of radiation-free methods, such as ultrasonography or methods with lower radiation dose than tomography, such as digital tomosynthesis, for chest imaging in the emergency department can be expanded. Further research is needed in this area in the near future.

**Author Contributions:** Conceptualization, C.A., E.S.B. and E.K.; methodology, C.A., E.S.B. and E.K.; software, C.A., E.S.B., E.K. and F.N.K.; validation, C.A., E.S.B. and E.K.; formal analysis, C.A. and F.N.K.; investigation, C.A., E.S.B., E.K. and F.N.K.; resources, C.A., E.S.B., E.K. and F.N.K.; data curation, C.A., E.K. and F.N.K.; writing—original draft preparation, C.A., E.S.B., E.K. and F.N.K.; writing—review and editing, C.A., E.S.B. and F.N.K.; visualization, C.A. and E.K.; supervision, C.A. and E.S.B.; project administration, C.A. All authors have read and agreed to the published version of the manuscript.

**Funding:** This research received no external funding.

**Institutional Review Board Statement:** The study was conducted by the Declaration of Helsinki and approved by the Non-Interventional Clinical Research Ethics Committee of İzmir Kâtip Çelebi University (Decision No: 0129, Date: 23 March 2023).

**Informed Consent Statement:** Since the study was designed retrospectively, no separate patient consent was obtained except for the ethics committee's approval.

**Data Availability Statement:** The data presented in this study are available on request from the corresponding author. The data are not publicly available due to [privacy and ethical].

**Conflicts of Interest:** The authors declare no conflict of interest.

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
