# Peer review of "Have Chest Imaging Habits Changed in the Emergency Department after the Pandemic?"

_tomography, doi:10.3390/tomography9060163_

Round 1

Reviewer 1 Report

Comments and Suggestions for Authors

Thank you very much for this manuscript

This manuscript report chest imaging examinations performed in the emergency department in the pre-pandemic, pandemic, and post-pandemic period.

I think the results are very interesting, particularly the lower age of the patients who underwent CT in the post-pandemic period.

If I look at figure number 3 it seems that in 2023 the % of CT examinations is not so different from the pre-pandemic period.

Do you have something to say about that?

Could you comment on this?

Do you think that this data will be similar to pre-pandemic period in the future?  

Instead, if I look at figure 2 it seems that the mean age of people who perform chest CT is permanently lower in the post-pandemic period with respect to pre-pandemic one.  

Can you explain how you decided that the pandemic period started in March 2020 and not before and particularly that post-pandemic started in April 2022?

Minor: 

Line 100: I suggest to change "gave their blessing for" into "approved"

I suggest to modify "applications" and "applyed" into "admissions" and "presented"

I suggest to modify "Thorax computed tomography (TCT)" into "Chest computed tomography (CCT)"

Thank you very much

Comments on the Quality of English Language

English needs minor revision

Line 100: I suggest to change "gave their blessing for" into "approved"

I suggest to modify "applications" and "applyed" into "admissions" and "presented"

I suggest top modify "Thorax computed tomography (TCT)" into "Chest computed tomography (CCT)"

Author Response

Revision Note for Reviewer 1

 Comments and Suggestions for Authors

Thank you very much for this manuscript. This manuscript report chest imaging examinations performed in the emergency department in the pre-pandemic, pandemic, and post-pandemic period. I think the results are very interesting, particularly the lower age of the patients who underwent CT in the post-pandemic period.

If I look at figure number 3 it seems that in 2023 the % of CT examinations is not so different from the pre-pandemic period.

Do you have something to say about that?

Could you comment on this?

Do you think that this data will be similar to pre-pandemic period in the future? Instead, if I look at figure 2 it seems that the mean age of people who perform chest CT is permanently lower in the post-pandemic period with respect to pre-pandemic one.

Can you explain how you decided that the pandemic period started in March 2020 and not before and particularly that post-pandemic started in April 2022?

Answer: Thank you very much for your valuable comments and suggestions. When Figure 3 is examined, there may be similarities between the data of the first 3 months of 2023 and the pre-pandemic period, but it would be more accurate to look at data covering the whole year, not just 3-month data. Considering that the data followed a fluctuating course during the pandemic period, looking only at data in short periods can be misleading.

However, the issue you touched upon is very important and at the same time reflects the main idea that this research wants to emphasize. We hope that these data will reach levels similar to those in the pre-pandemic period in the future, by disseminating and supporting the idea of 'taking care to use tomography in the correct indications and reducing radiation exposure', especially among emergency room physicians.

The time periods in the study were divided into pre-pandemic, pandemic and post-pandemic based on the number of cases in Turkey and the center where the research was conducted, the number of patients hospitalized and in intensive care, and the working order in the hospital. The first COVID-19 cases in Turkey and in the city where the research was conducted began to be seen in March 2020. In January and February 2020, COVID-19 suspected cases were evaluated in the "Infectious Diseases and Clinical Microbiology" department. However, with the increase in the number of COVID-19 suspected cases since March 2020, it was decided by the competent health authorities that the first admission of all COVID-19 suspected patients should be made to the 'Emergency Medicine Clinic'. As of April 2022, thanks to protective measures and vaccination, the COVID-19 epidemic was brought under control and the number of cases decreased in Turkey and in the hospital where the research was conducted, as well as all over the world. Emergency room visits and hospitalizations with suspected COVID-19 have almost completely ended. For this reason, COVID-19 suspected cases were primarily transferred to the "Infectious Diseases and Clinical Microbiology" department and the emergency medicine clinic returned to its normal functioning.

Minor: 

Line 100: I suggest to change "gave their blessing for" into "approved"

I suggest to modify "applications" and "applyed" into "admissions" and "presented"

I suggest to modify "Thorax computed tomography (TCT)" into "Chest computed tomography (CCT)"

Thank you very much.

Answer: Thank you very much for your suggestion. All edits have been made in line with your suggestions.

Comments on the Quality of English Language

English needs minor revision

Line 100: I suggest to change "gave their blessing for" into "approved"

I suggest to modify "applications" and "applyed" into "admissions" and "presented"

I suggest top modify "Thorax computed tomography (TCT)" into "Chest computed tomography (CCT)"

Answer: Thank you very much for your suggestion. All edits have been made in line with your suggestions.

Kind regards,

Reviewer 2 Report

Comments and Suggestions for Authors

This paper is a comparative study of the demographic characteristics of patients before and after the pandemic, considering the temporal conditions.

It is recommended to include not only a comparative study of patients who underwent Chest X-rays (CXR) and Chest Computed Tomography (CT) scans, but also information regarding the conditions that led to the imaging in order to make a comprehensive evaluation.

Author Response

Revision Note for Reviewer 2

 This paper is a comparative study of the demographic characteristics of patients before and after the pandemic, considering the temporal conditions.

It is recommended to include not only a comparative study of patients who underwent Chest X-rays (CXR) and Chest Computed Tomography (CT) scans, but also information regarding the conditions that led to the imaging in order to make a comprehensive evaluation.

Answer: Thank you very much for your valuable comments and suggestions. You are absolutely right in your suggestion and we agree with you. This suggestion is a guide for new studies that can be done in the future and should definitely be taken into consideration. However, considering the total number of patients in the study, unfortunately it was not possible to make a classification as you mentioned with our current workforce. We stated this as one of the limitations of the study, but we made arrangements in the limitations section to make it more understandable.

Kind regards,

Reviewer 3 Report

Comments and Suggestions for Authors

I read with interest the submitted article and I would like to thank the authors for their scientific work.

Line 97: In addition, the university is connected to the hospital in some way

I find his statement quite confusing. The writers should explain the process of referring patients to the tertiary hospital and mention if there is a screening process through primary care physicians. Additionally it should be clarified how this changed during COVID. Maybe there were more X-rays done in small clinics / private practices?  This section should be rewritten.

Line 131 : performed per 100 admissions…

The writers should clarify if the mean ED admissions or Hospital admissions throughout the text.

One important missing part in the analysis is the factor of the indication for the visit to the ED. It would be of added value if the writers could provide the percentage of ED visits that were performed because of an infectious reason : coughing, febrile patients etc. and include this in their analysis. In this way it could be demonstrated that despite the improvement of the pandemic and switch to a more conventional allocation of the ED diagnosis, the pattern of preferring CT to Rx remained. The authors mention it in their limitations but it seems to be a key element of proving their point. It would like to see at least a categorical distinction between infectious or not infectious if it is possible in the demographics.

Moreover it would be quite interesting to document the number of patients that got an Rx AND a CT and compare them with the percentage that had an CT as first option.

There is extensive literature regarding the value of lung ultrasound (LUS) in replacing Rx/CT for screening, diagnosing and following up patients with suspected COVID infection.  Was LUS implemented in this center? If not this should be at least mentioned in the discussion as another option – future direction.

Comments on the Quality of English Language

An edtiting of the English language by a native speaker would add to the overall presentation of this article and is strongly advised. 

For example : Ethics Committee gave their blessing for the study

Author Response

Revision Note for Reviewer 3

Comments and Suggestions for Authors

I read with interest the submitted article and I would like to thank the authors for their scientific work.

Line 97: In addition, the university is connected to the hospital in some way

I find his statement quite confusing. The writers should explain the process of referring patients to the tertiary hospital and mention if there is a screening process through primary care physicians. Additionally it should be clarified how this changed during COVID. Maybe there were more X-rays done in small clinics / private practices?  This section should be rewritten.

Answer: Thank you very much for your valuable comments and suggestions. We understand the reason for the confusion. First of all, we would like to point out that the hospital where the research was conducted is a tertiary level training and research hospital and has an experience of more than 40 years. In line with the health policies implemented in Türkiye, if a medical school (especially if it is newly opened) does not have a full-fledged hospital, it continues its clinical training in a training and research hospital with which it is affiliated. This is intended to be stated in this section. However, in line with your suggestion, we considered removing this statement to avoid confusion. Additionally, there is no mandatory referral chain in line with health policies in Türkiye. Patients can directly admitted to the emergency department of a primary, secondary or tertiary hospital at their own discretion. On the other hand, if necessary, they can be referred from primary and secondary hospitals to tertiary hospitals. The center where the research was conducted is located in the city center and there are only family health centers nearby. These primary care centers do not have imaging facilities. Referred patients from primary and secondary level hospitals are also admitted to the center where the research is conducted. However, lung imaging of these patients is usually performed in the hospital to which they first admitted, and since it is recorded in the electronic 'e-Nabız' health system, it can also be viewed in the institution to which they are referred.

Line 131 : performed per 100 admissions…

The writers should clarify if the mean ED admissions or Hospital admissions throughout the text.

Answer: Thank you for your suggestion. All rates were calculated solely on the number of patients applying to the emergency department. All edits have been made in line with your suggestions.

One important missing part in the analysis is the factor of the indication for the visit to the ED. It would be of added value if the writers could provide the percentage of ED visits that were performed because of an infectious reason : coughing, febrile patients etc. and include this in their analysis. In this way it could be demonstrated that despite the improvement of the pandemic and switch to a more conventional allocation of the ED diagnosis, the pattern of preferring CT to Rx remained. The authors mention it in their limitations but it seems to be a key element of proving their point. It would like to see at least a categorical distinction between infectious or not infectious if it is possible in the demographics.

Moreover it would be quite interesting to document the number of patients that got an Rx AND a CT and compare them with the percentage that had an CT as first option.

Answer: Thank you very much for your comment and suggestion. You are absolutely right in your suggestion and we agree with you. This suggestion is a guide for new studies that can be done in the future and should definitely be taken into consideration. However, considering the total number of patients in the study, unfortunately it was not possible to make a classification as you mentioned with our current workforce. Unfortunately, it was not possible to access this data from the hospital's electronic records. We stated this as one of the limitations of the study, but we made arrangements in the limitations section to make it more understandable.

There is extensive literature regarding the value of lung ultrasound (LUS) in replacing Rx/CT for screening, diagnosing and following up patients with suspected COVID infection.  Was LUS implemented in this center? If not this should be at least mentioned in the discussion as another option – future direction.

Answer: Thank you very much for your suggestion and valuable contribution. Lung ultrasound is a subject that the authors attach great importance to. In fact, in the past, a study on lung ultrasound was conducted in a center where this study was conducted. (Acar H, Yamanoğlu A, Arıkan C, Bilgin S, Akyol P, Kayalı A, Karakaya Z (2022). Effectiveness of the CLUE protocol in COVID-19 triage. Cukurova Medical Journal, 47(2), 722 - 728. 10.17826/cumj.1086062). However, due to the large number of patients available and the small number of specialist doctors competent in lung ultrasound, lung ultrasound was not routinely performed on patients. Lung ultrasound has been performed in only a limited number of selected critically ill patients. In line with your suggestion, the topic of lung ultrasound has been added to the discussion.

Comments on the Quality of English Language

An edtiting of the English language by a native speaker would add to the overall presentation of this article and is strongly advised. 

For example : Ethics Committee gave their blessing for the study

Answer:

Thank you very much for your valuable comments and suggestions. All edits have been made in line with your suggestions

Kind regards,

Round 2

Reviewer 2 Report

Comments and Suggestions for Authors

This paper is an analysis based on medical field experience, and as the author mentioned, it is a good study that needs to be expanded into multi-institutional joint research.

It was revised to fully reflect the reviewer's considerations.

Author Response

REVÄ°EWER 2

This paper is an analysis based on medical field experience, and as the author mentioned, it is a good study that needs to be expanded into multi-institutional joint research.

 It was revised to fully reflect the reviewer's considerations.

Response

Thank you for your supportive critics. It really servet o improve the quality of our research. We also add your last comment to our limitations.

Best regards

Reviewer 3 Report

Comments and Suggestions for Authors

I want to thank the authors for their effort to significanty improve the paper. Nevertheless i still have two major observations:

- It remains unclear if the x-rays that the patients had maybe already received before, in primary and secondary care centers, are included in the calculation of the preferred first radiologic imaging. If patients are referred to the tetartiary center with an imaging study , it makes sense that the ED doctors will choose the CT for further investigation. This creates an important bias in the study. This must be clearly written in the methods section. 

- The biggest problem remains the clinical stratification of the patient flow. I understand that access to the files could be challenging but maybe the indication for every imaging test could be found in the report.  A distinction between infectious and non-infectious would be enough. This is another big bias that dillutes the importance of the findings significantly.

Comments on the Quality of English Language

The language is improved. 

Author Response

RESPONSE TO REVÄ°EWER 3

I want to thank the authors for their effort to significanty improve the paper. Nevertheless i still have two major observations:

  • It remains unclear if the x-rays that the patients had maybe already received before, in primary and secondary care centers, are included in the calculation of the preferred first radiologic imaging. If patients are referred to the tetartiary center with an imaging study , it makes sense that the ED doctors will choose the CT for further investigation. This creates an important bias in the study. This must be clearly written in the methods section. 

Response 1

Dear reviewer,

First, we are glad to hear that our study is better after your constructive criticism.

For your first observation, the comment that you tell us is precious and critical. In our city and town, no hospital has computerizing technology; on the other hand, we have a national system that we can check if there are other radiological or laboratory results for the patients, what they are using like medicaments, etc and this is passing online to the web system. So, we avoid taking repetitive imaging studies in this hospital, and we already exclude the patients who have imaging before our hospital.

We add this important information in material method in exclusion criteria part.

‘’Patients admitted to the adult emergency department of the hospital between January 2019 and March 2023 were included in the study. Patients with missing data in their files and patients who were referred from another hospital and had imaging in that hospital were excluded from the study by checking the national health bank e-nabız (an application where citizens and healthcare professionals can access health data collected from healthcare institutions via internet and mobile devices), which can only be used by physicians.’’

  • The biggest problem remains the clinical stratification of the patient flow. I understand that access to the files could be challenging but maybe the indication for every imaging test could be found in the report.  A distinction between infectious and non-infectious would be enough. This is another big bias that dillutes the importance of the findings significantly.

Response 2

Dear reviewer

You are entirely right about this subject. When we designed this manuscript, we did not consider classification because the patients who came during the pandemic (viral patients, traumatic patients, all the indications that need CT) and post-pandemic were the same. We did not want to classify the indications because we aimed to find just the change of habits in using radiography in an emergency department that accepts similar patients yearly. We already added these details in the limitation part.

‘’ An additional constraint of the research was that, for the purposes of the study, patients admitted to the emergency department were not categorized based on clinical diagnoses, complaints, symptoms, or imaging indications.’’

Best regards,